# Modeling ETBF-Mediated Colorectal Tumorigenesis Using AOM/DSS in Wild-Type Mice

**DOI:** 10.3390/ijms26136218

**Published:** 2025-06-27

**Authors:** Soonjae Hwang, Yeram Lee, Ki-Jong Rhee

**Affiliations:** 1Department of Biochemistry, Lee Gil Ya Cancer and Diabetes Institute, College of Medicine, Gachon University, 155 Gaetbeol-ro, Yeonsu-gu, Incheon 21999, Republic of Korea; soonjae@gachon.ac.kr (S.H.); ramii25@gachon.ac.kr (Y.L.); 2Department of Health Sciences and Technology, GAIHST (Gachon Advanced Institute for Health Sciences & Technology), Gachon University, Incheon 21999, Republic of Korea; 3Department of Biomedical Laboratory Science, College of Software and Digital Healthcare Convergence, Yonsei University MIRAE Campus, Wonju 26493, Republic of Korea

**Keywords:** tumorigenesis, ETBF, BFT, inflammation, colitis, DSS

## Abstract

Enterotoxigenic *Bacteroides fragilis* (ETBF) promotes colitis-associated cancer through the *Bacteroides fragilis* toxin (BFT), which induces colonic inflammation that can be exacerbated by external stimuli. We found that BALB/c mice infected with ETBF and treated with azoxymethane and dextran sodium sulfate (DSS) developed numerous distal colon polyps more rapidly than B6 mice, suggesting strain differences in ETBF-induced tumorigenicity. Using a *bft* gene-deficient ETBF strain, we confirmed BFT’s crucial role in ETBF-promoted tumorigenesis and inflammation. While both 1% and 2% DSS induced comparable polyp formation, 1% DSS minimized mortality, proving sufficient for maximizing polyp development. Mechanistically, BFT-mediated tumorigenesis involves NF-κB/CXCL1 signaling in colonic epithelial cells exposed to BFT and DSS, a pathway known to be critical for inflammation and cancer progression. This model provides a valuable platform for dissecting ETBF’s colitis-associated cancer-promoting mechanisms, particularly those involving BFT, and for evaluating BFT-targeted therapeutic interventions.

## 1. Introduction

Colorectal cancer (CRC) is a leading cause of cancer-related mortality worldwide [1,2,3]. Chronic colonic inflammation is a key driver of tumor development [4,5,6], a phenomenon particularly evident in individuals with inflammatory bowel disease (IBD), who exhibit a substantially increased risk of CRC [7,8]. The gut microbiota is recognized as a critical environmental factor influencing intestinal inflammation [9,10,11]. Consequently, significant research efforts have focused on identifying specific bacterial species contributing to colitis and/or colonic tumorigenesis and elucidating the mechanisms by which these microbes modulate inflammatory responses [12].

Enterotoxigenic *Bacteroides fragilis* (ETBF), a commensal bacterium of the human gut, has been demonstrated to induce colonic inflammation in murine models [13,14,15]. The pathogenic potential of ETBF is primarily attributed to the production of *Bacteroides fragilis* toxin (BFT), an exotoxin that disrupts the integrity of epithelial tight junctions [16], subsequently triggering an inflammatory immune response within the large intestine [15]. The observed increased prevalence of ETBF in individuals with CRC compared to healthy controls [17,18,19] has generated considerable interest in elucidating the mechanisms by which BFT contributes to tumorigenesis. Studies using animal models of ETBF-associated carcinogenesis have revealed that BFT exacerbates tumor development by promoting the production of pro-inflammatory cytokines, including interleukin-17A (IL-17A) and CXCL1, in the Min^Apc+/−^ mouse model [20,21,22]. Mechanistically, BFT, a zinc-dependent metalloprotease, targets and cleaves E-cadherin [23], thereby compromising the intestinal epithelial barrier and activating NF-κB signaling pathways in colonic epithelial cells [24,25,26,27]. Furthermore, BFT exposure induces CXCL1 expression and secretion by colonic epithelial cells via NF-κB signaling [28].

The Min^Apc+/−^ mouse model and the AOM/DSS model are two prominent models employed to investigate the mechanisms of ETBF-driven CRC development [29,30]. Our group has previously demonstrated that ETBF infection enhances colonic tumorigenesis in the AOM/DSS model [31]. However, important unresolved issues remain, including the influence of mouse strain on ETBF-induced tumorigenesis, the optimization of DSS cycling and concentration for efficient colonic polyp induction, and the lack of well-defined in vivo systems to comprehensively investigate the mechanisms by which ETBF promotes colitis-associated cancer. The AOM/DSS model, designed to mimic human colitis-associated CRC [32], combines AOM treatment with DSS administration. AOM undergoes metabolic activation via CYP2E1, which catalyzes hydroxylation at the methyl group distal to the N(O) function, converting AOM into methylazoxymethanol (MAM) [33,34], a highly reactive alkylating agent that generates 6-O-methylguanine adducts in DNA, ultimately leading to mutation accumulation and carcinogenesis. DSS, a sulfated polysaccharide, induces colonic epithelial damage, increased mucosal permeability, and transmural inflammation in mice [35,36]. This widely used model is considered robust and reproducible for studying inflammation-associated colorectal carcinogenesis in wild-type rodents [37,38,39]. Typical AOM/DSS protocols involve one or more AOM injections followed by repeated cycles of 2–3% (*w*/*v*) DSS in drinking water to induce colitis [40,41]. However, the dosages of AOM (5–12.5 mg/kg) and DSS (1–3%) vary considerably across studies [42,43,44], and a standardized, time-course-dependent protocol for the AOM/DSS model combined with ETBF infection is lacking. Polyp formation and the risk of mortality due to excessive colitis in the AOM/DSS model are dependent on the DSS dosage, frequency of administration, and experimental duration.

In this study, we evaluated an ETBF/AOM/DSS model using a single AOM injection combined with one or three cycles of 2% DSS to assess tumorigenicity in the colons of BALB/c and C57BL/6 mice. We also investigated the tumorigenic potential of the *bft* gene in ETBF-mediated tumorigenesis within this AOM/DSS model. To optimize the DSS concentration for reliable polyp induction, we assessed several key parameters in mice exposed to either 1% or 2% DSS, including polyp number and size, overall survival rate, and levels of systemic inflammatory markers. We found that a reduced DSS concentration (1%) in the ETBF/AOM/DSS model reliably induces polyp formation, mimicking the effects of ETBF infection, without excessive mortality or variability in polyp development. In vitro, we examined the effects of BFT and/or DSS on IL-8 expression and NF-κB activity in colonic epithelial cells. Establishing a robust and standardized ETBF/AOM/DSS model is crucial for dissecting the complex interplay between ETBF infection, inflammation, and CRC development, ultimately paving the way for targeted therapeutic strategies. Evaluation of these endpoints allowed us to refine the ETBF/AOM/DSS model and establish a standardized protocol. This refined model will serve as a valuable tool for future investigations into the intricate interplay between ETBF, inflammation, and CRC.

## 2. Results

### 2.1. Susceptibility to Polyp Formation in ETBF/AOM/DSS-Treated C57BL/6 and BALB/C Mice

To compare the susceptibility of C57BL/6 and BALB/c mouse strains to ETBF/AOM/DSS treatment, mice were injected once with AOM (10 mg/kg) and then given drinking water containing clindamycin and gentamicin for 12 days, starting 2 days post-injection (Figure 1A). Wild-type enterotoxigenic *Bacteroides fragilis* (WT-ETBF; *bft-2*) bacterial strain was orally inoculated at day 7. At day 21, the first DSS cycle (5 days of 1% DSS + 16 days of distilled water) was initiated with a total of three DSS cycles. After the third DSS cycle, the colon was examined macroscopically. C57BL/6 mice showed a median of 14 polyps, whereas BALB/c mice showed a median of 36 polyps (Figure 1B,C). The median polyp size was also significantly greater in BALB/c mice compared to C57BL/6 mice (Figure 1D). These results demonstrate that BALB/c mice exhibit a greater polyp burden and polyp size compared to C57BL/6 mice in the ETBF/AOM/DSS model. Henceforth, all subsequent experiments were conducted using the BALB/c mouse strain.

### 2.2. Tumorigenicity of ETBF-Colonized Mice Treated with AOM/DSS

While previous work has shown that WT-ETBF infection exacerbates AOM/DSS-induced tumorigenesis [31], the influence of DSS cycle number on polyp development and size in this model remains limited. Therefore, we investigated the rapidity of polyp formation in WT-ETBF (*bft-2*; E2) pre-infected mice [AOM/DSS + WT-ETBF (*bft-2*), hereafter referred to as A/D + E2] by assessing polyp burden and size after one and three DSS cycles (Figure 1B and Figure 2A). Notably, no polyp formation was observed in the AOM/DSS control group after a single DSS cycle. However, all mice in the A/D + E2 group developed polyps after a single DSS cycle (Figure 2D,E). After the third DSS cycle, the AOM/DSS control group developed a median of 9 polyps per mouse, whereas the A/D + E2 group developed a median of 37 polyps, representing a 4-fold increase (Figure 2E). Furthermore, polyp size was also significantly greater in the A/D + E2 group compared to the control group (Figure 2F). These data further substantiate the observation that ETBF infection accelerates polyp development in the context of AOM/DSS-induced tumorigenesis. Importantly, these findings show that even a single DSS cycle is sufficient to induce detectable polyp formation in the AOM/DSS model applied to ETBF infection.

### 2.3. ETBF-Accelerated Tumorigenesis Is Driven Exclusively by the Bft Gene

ETBF strains encode two distinct secretory metalloproteinases, BFT and MPII, within their pathogenicity island [45,46,47], but while BFT’s role in colitis and tumorigenesis is well-established, MPII’s contribution remains less studied. Previous research using a recombinant ETBF strain harboring the *bft* gene in a non-toxigenic *B. fragilis* (NTBF) strain showed increased polyp formation in the AOM/DSS model compared to the NTBF control group [31], but this increase was less pronounced than that observed with WT-ETBF in the AOM/DSS model. To definitively determine whether BFT is solely responsible for ETBF-mediated effects in the AOM/DSS system, we utilized a *bft-1* gene-deleted ETBF strain (E1 Δ*bft-1*) and assessed its impact on tumor formation and survival (Figure 3A). While the WT-ETBF (*bft-1*)-infected AOM/DSS group exhibited robust polyp formation (Figure 3B,C), the AOM/DSS group infected with the *bft-1* deletion ETBF strain showed a marked decrease in polyp development comparable to the AOM/DSS group (Figure 3B,C). The survival rate in the *bft-1* deletion ETBF-infected group remained unchanged at 100%, comparable to the AOM/DSS control group. In contrast, the WT-ETBF-infected group exhibited a significantly lower survival rate (63.6%) (Figure 3D). These findings confirm that the *bft* gene is the primary virulence factor driving ETBF-accelerated polyp formation in this model.

### 2.4. Bft-1 Is Essential for Systemic Inflammation in the ETBF/AOM/DSS Model

Having confirmed the *bft* dependence of ETBF-induced tumorigenesis (Figure 3), we next investigated its role in systemic inflammation in mice. Comparable colonization of ETBF was observed in AOM/DSS-treated mice infected with ETBF(Δ*bft-1*) or ETBF (*bft-1*) (Appendix A). We measured established markers of systemic inflammation, including spleen weight, colon length, and colon weight. In the WT-ETBF (*bft-1*)-infected AOM/DSS group, we observed increased spleen weight, decreased colon length, and increased colon weight compared to the AOM/DSS control group (Figure 4A–C). Consistent with these observations, serum levels of CXCL1, IL-17A, and nitric oxide were also elevated in the WT-ETBF-infected group (Figure 4D–F). In contrast, the AOM/DSS group infected with the *bft-1*-deleted ETBF strain (Δ*bft-1*) exhibited reduced spleen weight, increased colon length, and a decreased ratio of colon weight to colon length (Figure 4A–C). Similarly, serum levels of CXCL1, IL-17A, and nitric oxide were also reduced in this group (Figure 4D–F).

### 2.5. Quantitative Assessment of the Impact of DSS Dosage on ETBF-Promoted Tumorigenesis

In the AOM/DSS model, AOM initiates tumorigenesis, while DSS induces intestinal inflammation, with the severity of inflammation and subsequent polyp formation positively correlating with DSS concentration [40]. To determine the optimal DSS concentration for an ETBF-induced polyp formation model within this framework, we compared the effects of 1% and 2% DSS on polyp development, survival rate, and circulating levels of IL-17A, CXCL1, and nitric oxide. DSS dosage did not influence ETBF colonization in AOM/DSS-treated mice (Appendix A). Polyp incidence was comparable between the two groups (Figure 5B,C). However, the average polyp volume in the ETBF/AOM/DSS (2%) group was significantly greater than that in the 1% DSS group (Figure 5D). Conversely, the survival rate in the 2% DSS group was markedly lower (66.7%) compared to the 1% DSS group, which exhibited full survival (100%) (Figure 5E). Serum levels of IL-17A, CXCL1, and nitric oxide were all significantly higher in the 2% DSS group (Figure 5F–H). These results suggest that 1% DSS is sufficient for establishing an ETBF-induced polyp model in the AOM/DSS system. This concentration yields a comparable tumor incidence to 2% DSS without compromising the survival rate. Furthermore, the 1% DSS concentration may offer advantages such as reduced variability in polyp development and economic benefits due to reduced DSS consumption.

### 2.6. Co-Treatment with BFT and DSS Elevates NF-κB Luciferase Activity and IL-8 Expression in Intestinal Epithelial Cells

To investigate the synergistic inflammatory effects of DSS and BFT in our tumorigenesis model, we examined BFT-induced signaling in HT29/C1 human colonic carcinoma cells. BFT induces cell rounding, dependent on E-cadherin cleavage and leading to NF-κB activation and IL-8 expression in HT29/C1 cells [24,26,28]. Initially, we performed a trypan blue assay to confirm that the combination of DSS and rET (the culture supernatant of recombinant *B. fragilis* secreting active BFT) did not directly decrease cell viability (Figure 6A), thus ruling out a cytotoxic effect as the cause of increased inflammation. Having established this, we next investigated the impact of DSS on BFT-induced cell rounding. Co-treatment with 1% DSS did not prevent rET-induced cell rounding (Figure 6B). To gain further mechanistic insight, we examined IL-8 (the human equivalent of mouse CXCL1) expression and NF-κB luciferase activity. rET and DSS co-treatment significantly increased both IL-8 expression and NF-κB luciferase activity compared to rET or DSS alone (Figure 6C,D). The addition of an NF-κB inhibitor (BAY 11-7082; 10 μM) to BFT- and DSS-treated HT29/C1 cells reduced IL-8 expression and NF-*κ*B reporter activity (Figure 6C,D). These results demonstrate that co-treatment with DSS and BFT elevates NF-κB/CXCL1 signaling in colonic epithelial cells, suggesting a synergistic inflammatory mechanism.

## 3. Discussion

CRC is a leading cause of cancer-related mortality in developed nations worldwide. The gut microbiota plays a critical role in the development and progression of this malignancy [1,2,3,7,9]. CRC typically arises from benign adenomas within the colon before progressing to malignancy [48]. Considerable research is focused on elucidating the mechanisms driving this adenoma-to-carcinoma transition, particularly the influence of the gut microbiome [49]. Among the various gut bacteria implicated, the role of ETBF in CRC development and progression has been a subject of intense scientific inquiry.

In the present study, BALB/c mice exhibited greater susceptibility to the ETBF/AOM/DSS regimen compared with C57BL/6 mice. This observation aligns with previous reports demonstrating a heightened susceptibility of BALB/c mice to polyp formation following AOM administration alone, while C57BL/6 mice show a lower incidence [42]. Thus, the increased polyp formation observed in BALB/c mice treated with ETBF/AOM/DSS may be attributed, at least in part, to their inherent sensitivity to AOM. In our research group, we have used both the BALB/c and C57BL/6 mouse strains to investigate the effects of ETBF/AOM/DSS on colorectal carcinogenesis. We first reported the tumor-promoting potential of ETBF in the AOM/DSS model using the BALB/c strain [31]. The BALB/c strain was used to investigate the effects of zerumbone in the ETBF/AOM/DSS model [50]. In other subsequent studies, we used the C57BL/6 strain to examine the effects of natural product extracts and dietary salt in the ETBF (*bft-2*)/AOM/DSS model [51,52]. The selection of mouse strains was initially based on the strains used by other researchers for the respective studies. In the current study, we performed a side-by-side comparison of BALB/c and C57BL/6 strains. Future studies are warranted to investigate the susceptibility of other inbred mouse strains (e.g., 129S1/Sv, A/J, AKR/J, C3H/HeJ) to the ETBF/AOM/DSS model.

The experimental findings consistently demonstrate a significant strain-specific susceptibility to azoxymethane/dextran sulfate sodium (AOM/DSS)-induced colorectal polyp development in mice. In studies by Suzuki et al. [42] and Schepelmann et al. [53], BALB/c mice invariably exhibited a higher incidence of polyp formation, reaching 100% in both investigations. Notably, the Suzuki et al. [42] study further quantified this difference, reporting a considerably greater average number of polyps in BALB/c mice (11.4 ± 5.9) compared to C57BL/6 mice (2.5 ± 2.1), which showed a lower polyp incidence (80% or 70%). This striking differential response highlights the critical role of genetic background in modulating susceptibility to chemically induced colorectal carcinogenesis. The heightened sensitivity observed in BALB/c mice suggests the presence of specific genetic predispositions that may either facilitate the initiation and progression of AOM/DSS-induced lesions or impair endogenous protective mechanisms against inflammatory responses. Conversely, the relative resistance of C57BL/6 mice indicates potential genetic factors that confer resilience to the AOM/DSS regimen. Understanding these strain-specific differences provides valuable insights into the complex interplay between genetic factors and environmental insults in colorectal cancer pathogenesis and can guide the selection of appropriate mouse models for future research.

This study further explores how ETBF influences tumor formation within the AOM/DSS model of colitis-related cancer. Our results indicate that ETBF colonization amplifies DSS-driven inflammation, primarily through BFT activity. Specifically, the combination of BFT and DSS heightened NF-κB transcriptional activation, leading to increased CXCL1 expression, a well-established NF-κB target gene [54], in colonic epithelial cells. Previous studies have demonstrated that BFT-induced CXCL1 production facilitates myeloid cell infiltration into tumor sites, accelerating polyp formation, particularly in the distal colon [24,55].

Additionally, research has shown that ETBF infection prompts IL-17A secretion in the colonic mucosa [21,22]. This inflammatory cytokine plays a key role in activating the NF-κB pathway within intestinal epithelial cells, driving CXCL cytokine upregulation and promoting immune cell recruitment, which fosters tumor growth [24]. Within this process, the immune response of the host—particularly IL-17-mediated inflammation—works in tandem with BFT-driven NF-κB activation, collectively enhancing tumorigenesis.

To model ETBF-induced CRC development, the present study utilized the AOM/DSS system. In this model, AOM, a chemical carcinogen, initiates tumorigenesis by inducing DNA mutations in colonic epithelial cells. DSS, a chemical tumor promoter, disrupts intestinal mucosal epithelial tight junctions [40,41]. Previous studies have shown that ETBF infection alone in AOM-treated mice does not induce colonic polyp formation [31], contrasting with observations in Min^Apc+/−^ mouse models [22,56]. This discrepancy may be explained by the lower frequency of DNA mutations induced by AOM compared to genetically modified models. Furthermore, AOM-induced DNA damage may be repaired. Additionally, the inflammatory response triggered by ETBF alone may be less robust and sustained than that induced by DSS. Indeed, studies indicate that the IL-17 response in the colonic mucosa peaks 7–10 days after ETBF infection and subsequently declines [57]. In contrast, DSS-induced inflammation can be repeatedly induced and sustained by cyclical DSS administration, potentially resulting in a greater cumulative inflammatory burden compared to ETBF infection alone.

Our current study demonstrates that combined exposure of intestinal epithelial cells to BFT and DSS leads to increased IL-8 expression and NF-κB activity. This suggests that BFT and DSS synergistically activate the NF-κB/CXCL1 signaling pathway in colonic epithelial cells, thereby promoting tumorigenesis (Figure 7). This synergistic effect is a novel finding of the present study. While DSS is a synthetic chemical not typically encountered by humans, it can be considered a surrogate for environmental factors that induce inflammation in clinical settings. Although BFT secreted by ETBF may not, under normal circumstances, induce sufficient inflammation to directly cause tumors, it may contribute to polyp formation and CRC development in conjunction with other pro-inflammatory environmental factors. Further epidemiological and mechanistic studies are warranted to investigate this possibility.

Another potential mechanism for the increased tumor formation observed with ETBF and DSS could involve inflammation mediated by disruption of the intestinal barrier caused by BFT and DSS, leading to bacterial translocation. BFT is known to cleave E-cadherin, an adherence junction protein [16]. DSS has been shown to decrease the expression of ZO-1 and claudin proteins, components of tight junctions [58]. The rapid disruption of intestinal epithelial cell junctions caused by BFT and DSS may amplify the inflammatory response in the intestinal mucosa due to increased permeability and influx of microbial-derived endotoxins. This amplified inflammation may promote the survival and proliferation of microscopic tumor foci through IL-6/STAT3 activation, leading to rapid polyp development. Further in vitro and in vivo studies are needed to investigate this hypothesis.

ETBF harbors both the *bft* and *mpII* genes within its pathogenicity island [46,47]. Like BFT, MPII is a metalloproteinase with approximately 25% structural similarity to BFT [59]. Although the target proteins of MPII remain to be fully identified, the current study indirectly investigated its role in ETBF-induced CRC development using a *bft* deletion mutant of the ETBF strain. Consistent with other ETBF studies [22,60], our results confirmed that the tumor-promoting effect was abolished in the *bft*-deficient ETBF strain (Figure 3). These findings further support the critical role of BFT in ETBF-mediated CRC promotion. Future research should investigate the specific cell signaling pathways and gene expression changes induced by co-administration of purified BFT and MPII to intestinal epithelial cells compared to BFT alone. A deeper understanding of the distinct and synergistic effects of these two ETBF-secreted toxins could provide valuable insights into the mechanisms of ETBF-induced CRC and facilitate the development of targeted therapeutic strategies.

The gut microbiota plays a crucial role in maintaining host health by regulating digestion, nutrient absorption, immune responses, and defense against pathogens [61,62,63,64]. In experimental models such as AOM/DSS-induced colitis, microbial composition influences the severity of inflammation and tumor development [65,66], emphasizing its significance in disease progression. Given this impact, gut microbiota should be a key consideration in mechanistic studies on carcinogenesis, particularly in ETBF-mediated colorectal polyp formation, where microbial interactions may shape disease outcomes. Exploring the microbiome’s role in ETBF-induced tumorigenesis could deepen our understanding of microbial contributions to cancer progression and support the development of microbiota-targeted therapeutic approaches. Advancing knowledge in this field will strengthen colorectal cancer research, highlighting the microbiome’s relevance in both experimental and clinical studies.

The ETBF/AOM/DSS colorectal polyp model presented here serves as a valuable platform for investigating the complex interplay between ETBF and other environmental factors in CRC development. This model can also be utilized as a wild-type mouse model of CRC to dissect the underlying mechanisms of ETBF-induced carcinogenesis. Furthermore, its ability to induce a high number of polyps in a relatively short period makes it a useful tool for screening candidate anti-cancer and anti-inflammatory agents, as well as for general oncological research.

## 4. Materials and Methods

### 4.1. Bacterial Strains

This study utilized a collection of enterotoxigenic *B. fragilis* (ETBF) strains for infection: *B. fragilis* VPI 13784 (*bft*-1), *B. fragilis* VPI 13784 (Δ*bft*-1), and *B. fragilis* 86-5443-2-2 (*bft*-2). *B. fragilis* VPI 13784 (Δ*bft*-1) is a mutant strain derived from *B. fragilis* VPI 13784 (*bft*-1), with a targeted deletion of the *bft-1* gene. All wild-type *Bacteroides* strains included in this study exhibit inherent resistance to gentamicin. Consequently, all strains utilized in this study display clindamycin resistance, either through inherent mechanisms or via plasmid-mediated resistance conferred by pFD340. The bacterial strains were generously provided by Cynthia Sears and Augusto Franco-Mora (Johns Hopkins University, Baltimore, MD, USA).

### 4.2. Mouse Experiments

All animal care and experimental protocols were subject to rigorous ethical review and approval by the Institutional Animal Care and Use Committee of Yonsei University MIRAE campus (YWC-151005-1, YWCI-201612-014-01) and the Institutional Biosafety Committee of Yonsei University MIRAE campus (201612-P-014-01). All experimental procedures were conducted in strict adherence to the relevant guidelines and regulations established by these committees. Eight-week-old female BALB/c mice or C57BL/6 mice (obtained from Raon Bio, Yongin-si, Republic of Korea) were administered a single intraperitoneal injection of azoxymethane (AOM; Sigma, St. Louis, MO, USA) at a dose of 10 mg/kg body weight. Commencing two days post-AOM injection, mice received drinking water supplemented with clindamycin (100 mg/L) and gentamicin (300 mg/L) for a duration of five days to facilitate the establishment of *B. fragilis* colonization within the gastrointestinal tract. Following this antibiotic regimen, mice were orally inoculated with the bacterial strains of interest. The administration of antibiotic-supplemented water continued for an additional seven days. Subsequently, a cyclical regimen of dextran sodium sulfate (DSS) administration was initiated. Each cycle comprised five days of DSS treatment (36–50 kDa; MP Biomedicals, Santa Ana, CA, USA) followed by a 16-day recovery period with distilled water (DW). This cyclical regimen was repeated for a total of three cycles. For polyp counting, mice were sacrificed without prior knowledge of their experimental group assignments, ensuring an unbiased evaluation. The colorectal polyps were then directly counted to minimize potential observer bias. Azoxymethane (AOM) was procured from Sigma-Aldrich. Bacterial cultures were propagated in brain heart infusion broth and standardized to a concentration of 1 × 10^9^ colony-forming units (CFU) per 200 μL prior to oral inoculation of the mice. Bacterial colonization within the gastrointestinal tract was assessed through serial dilution and plating of fecal samples on brain heart infusion agar plates supplemented with gentamicin (50 μg/mL; Glentham Life Sciences, Corsham, UK) and clindamycin (6 μg/mL; Tokyo Chemical Industry Co., LTD., Tokyo, Japan). Characteristic *B. fragilis* colonies were enumerated following anaerobic incubation and expressed as CFU per gram of stool.

### 4.3. ELISA

Hemolysis-free sera were obtained from BALB/c mice colonized with ETBF and administered CAPE (Tokyo Chemical Industry, Tokyo, Japan) via cardiac puncture. Following coagulation at 4 °C, blood from each mouse was subjected to centrifugation (12,000× *g*, 4 °C, 20 min). The supernatant of each blood sample was carefully transferred to an autoclaved microcentrifuge tube and stored at −80 °C until further analysis. To quantify mouse CXCL1 and IL-17A levels, an enzyme-linked immunosorbent assay (ELISA) kit (R&D Systems, Minneapolis, MN, USA) was employed, strictly adhering to the manufacturer’s protocol.

### 4.4. qPCR Analysis of CXCL1 Expression in HT29/C1 Cells

Total RNA was extracted from HT29/C1 cells following the TRIZOL reagent extraction protocol (Invitrogen, Burlington, ON, Canada). Subsequently, 4000 ng of the extracted RNA served as a template for cDNA synthesis, which was carried out using the High-Capacity cDNA Reverse Transcription Kit (Invitrogen, Carlsbad, CA, USA). Quantitative PCR (qPCR) was then performed employing primers specific to both GAPDH (Thermo Fisher Scientific, Carlsbad, CA, USA) and CXCL1 (Thermo Fisher Scientific, Carlsbad, CA, USA). The resulting qPCR data for CXCL1 was normalized to the expression levels of the housekeeping gene GAPDH, consistent with previously established methodology [51].

### 4.5. Cell Viability Assay

HT29/C1 cells were seeded in 24-well plates and exposed to varying concentrations of either *B. fragilis* culture supernatants or dextran sulfate sodium (DSS; MP Biomedicals, Santa Ana, CA, USA). Following a 24 h incubation period, the supernatants were removed, and the cells were washed with phosphate-buffered saline (PBS; Life Technologies, Carlsbad, CA, USA). Subsequently, the cells were detached from the plate using 0.25% trypsin/EDTA (Life Technologies, Carlsbad, CA, USA) for 5 min at 37 °C. The detached cells were then resuspended in Dulbecco’s modified Eagle’s medium (DMEM; Life Technologies, Carlsbad, CA, USA) supplemented with 10% fetal bovine serum (FBS) to neutralize the trypsin (Life Technologies, Carlsbad, CA, USA) and halt its enzymatic activity. Cell viability was subsequently assessed using a Trypan blue exclusion assay (0.4%; Life Technologies, Carlsbad, CA, USA), following established methodology [51].

### 4.6. NF-κB Luciferase Reporter Assay

HT29/C1 cells were seeded in 24-well plates and incubated for 24 h prior to transfection. Transfection of reporter constructs, specifically NF-κB luciferase and *Renilla* luciferase plasmids, was performed using Lipofectamine^®^ 3000 (Invitrogen, Carlsbad, CA, USA) following the manufacturer’s protocol. Forty-eight hours post-transfection, cells were treated with either *B. fragilis* toxin (BFT) supernatant, dextran sulfate sodium (DSS; MP Biomedicals, Santa Ana, CA, USA), or BAY 11-7085 (an NF-κB inhibitor; Calbiochem, San Diego, CA, USA) for an additional 24 h period. Luciferase activity in the HT29/C1 cells was subsequently measured using the Dual-Luciferase^®^ Reporter Assay System (Promega, Madison, WI, USA). Luminescence was detected using a GloMax^®^ 20/20 luminometer (Promega, Madison, WI, USA), and firefly luciferase activity was normalized against *Renilla* luciferase activity to control for variations in transfection efficiency.

### 4.7. Treatment of Cells with Bacteroides Fragilis Toxin and Dextran Sulfate Sodium

Sterilization of *B. fragilis* culture supernatants was achieved using a 0.45 μm syringe filter (Merck, Rahway, NJ, USA) to eliminate bacterial cells, as previously described [51]. The resulting sterile supernatants were then stored at −80 °C. HT29/C1 cells were subsequently treated with culture supernatants derived from recombinant *B. fragilis* strains. These strains included NCTC9343 (rETBF) harboring pFD340::P-bft, which expresses and secretes wild-type BFT-2, and NCTC9343 (rNTBF) harboring pFD340::P-bftΔH352Y, which expresses and secretes a mutated, biologically inactive form of BFT. Prior to treatment, the culture media of the HT29/C1 cells were replaced with serum-free media following a PBS wash to mitigate potential BFT neutralization by serum proteins. The supernatants from each recombinant *B. fragilis* strain were diluted 1:10 in serum-free media. In select experiments, HT29/C1 cells were also treated with 1% *w*/*v* dextran sulfate sodium (DSS; MP Biomedicals, Santa Ana, CA, USA) and BAY 11-7085 (an NF-κB inhibitor; Calbiochem, San Diego, CA, USA) concurrently with the *B. fragilis* recombinant strain culture supernatants. Unless otherwise specified, all culture media and chemical reagents were obtained from GIBCO Life Technologies (Rockville, MD, USA).

### 4.8. Data Analysis

Statistical significance was assessed using GraphPad Prism 8 software (La Jolla, CA, USA), with a *p*-value of less than 0.05 considered statistically significant. Data are presented as median values across experimental groups. To compare differences between the two groups, a nonparametric Mann–Whitney U test was employed.

## 5. Conclusions

This study establishes a novel ETBF/AOM/DSS model in wild-type mice, demonstrating that ETBF infection promotes AOM/DSS-mediated tumorigenesis, particularly in BALB/c mice, via the action of BFT. Our findings reveal that BFT and DSS synergistically activate the NF-κB/CXCL1 signaling pathway in colonic epithelial cells, contributing to this enhanced tumorigenesis. Importantly, we optimized the DSS administration protocol, demonstrating that 1% DSS is sufficient for robust polyp induction in the ETBF/AOM/DSS model while minimizing mortality. This optimized and standardized ETBF/AOM/DSS model provides a valuable platform for dissecting the intricate interplay between ETBF infection, BFT activity, inflammation, and other contributing factors in colitis-associated cancer development. Furthermore, this model offers a powerful tool for evaluating potential therapeutic interventions targeting BFT and related pathways.

## Figures and Tables

**Figure 1 ijms-26-06218-f001:**
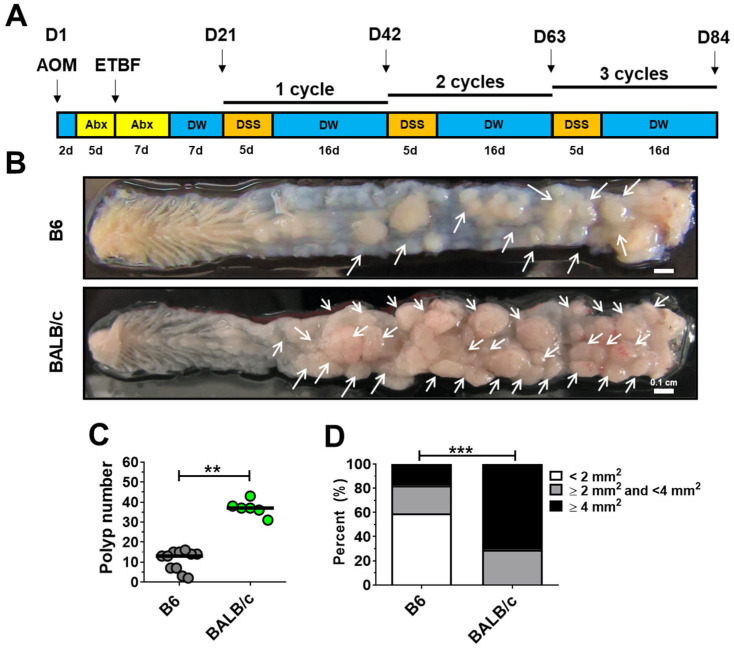
Mice strain susceptibility to the ETBF/AOM/DSS protocol. (**A**) ETBF/AOM/DSS protocol. C57BL/6 and BALB/c mice were given a single intraperitoneal injection of AOM (10 mg/kg) and provided with drinking water containing clindamycin/gentamicin for 5 days. WT-ETBF (*bft-2*; 1 × 10^9^ CFU) were orally inoculated, and the antibiotic cocktail continued for an additional 7 days. Seven days later, mice were subjected to three cycles of water and 2% DSS treatment. The total experimental period was 12 weeks. (**B**) Gross image of colon in ETBF/AOM/DSS-treated C57BL/6 (top panel) and BALB/c mice (lower panel) after 12 weeks. The distal colon is positioned to the right. White arrows indicate polyps. (**C**) Polyp number at 12 weeks. Each dot represents one mouse. Horizontal bar, median. (**D**) Polyp size distribution (*n* = 6–11 mice per group). Distribution of polyp sizes. Statistical significance was determined using the Mann–Whitney test. *** p* < 0.01, **** p* < 0.001.

**Figure 2 ijms-26-06218-f002:**
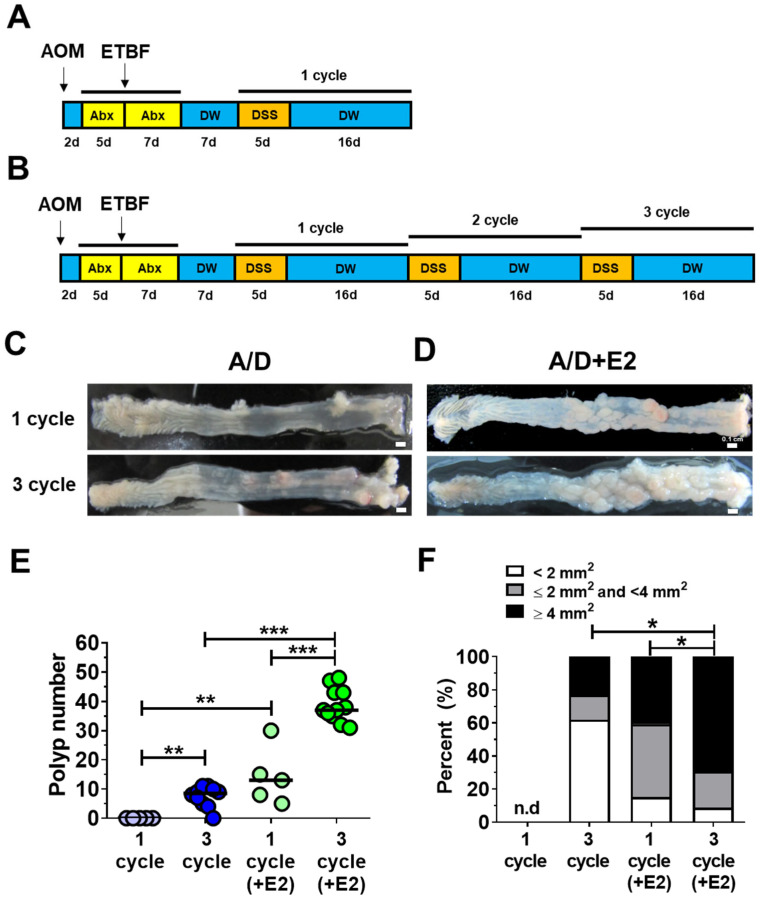
ETBF infection promoted AOM/DSS-induced tumorigenesis. Female BALB/c mice (8 weeks old) were administered a single intraperitoneal injection of AOM (10 mg/kg). Five days later, mice received drinking water containing clindamycin and gentamicin ad libitum for 5 days. Following this, mice were orally administered WT-ETBF (*bft-1*), and the antibiotic regimen was continued for an additional 7 days. Seven days after ETBF administration, mice underwent one or two cycles of 2% DSS in drinking water (5 days per cycle), followed by distilled water (DW) for 16 days per cycle. Impact of ETBF on AOM/DSS-induced colon tumorigenesis: (**A**) one DSS cycle; (**B**) three DSS cycles. (**C**) Gross image of colon in control BALB/c mice given AOM/DSS. (**D**) Gross image of colon in ETBF-colonized BALB/c mice given AOM/DSS. (**E**) Polyp number in BALB/c mice given AOM/DSS. Each dot represents one mouse (*n* = 5–10 mice per group). Horizontal bar, median. (**F**) Polyp size distribution. Distribution of polyp sizes. Statistical significance was determined using the Mann–Whitney test. * *p* < 0.05, ** *p* < 0.01, *** *p* < 0.001. n.d, not detected.

**Figure 3 ijms-26-06218-f003:**
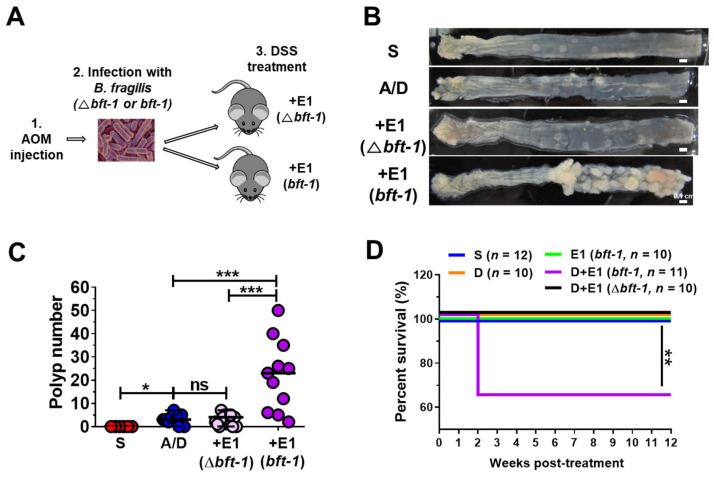
The *bft* gene is necessary for ETBF-promoted AOM/DSS tumorigenesis. AOM-treated BALB/c mice were infected with WT-ETBF (*bft-1*) or WT-ETBF (Δ*bft-1*) and subjected to three cycles of DSS (1%) for 12 weeks. (**A**) Schematic illustration of experimental design. (**B**) Representative gross macroscopic image of the colon. (**C**) Polyp number. Each dot represents one mouse (*n* = 6–17 mice per group). Horizontal bar, median. (**D**) Survival curve of mice. Kaplan–Meier curves depicting survival following ETBF infection in the AOM/DSS model, Mantel–Cox log-rank test. S, Sham; A/D, AOM/DSS alone; +E1, AOM/DSS + WT-ETBF (*bft-1*). * *p* < 0.05, ** *p* < 0.01, *** *p* < 0.001. ns, no statistical significance.

**Figure 4 ijms-26-06218-f004:**
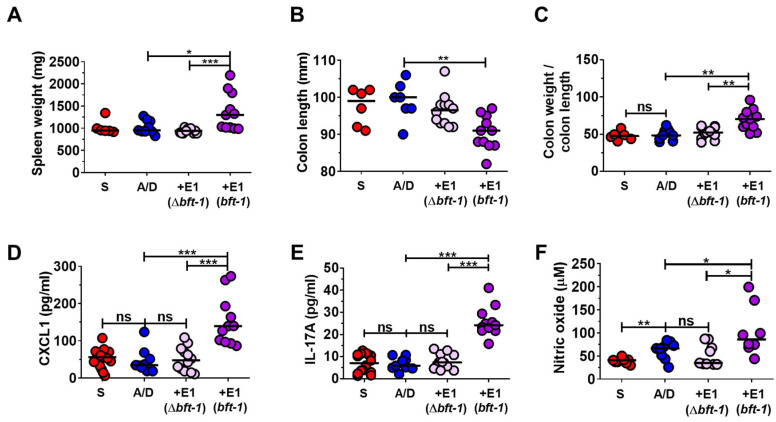
*bft-1* is essential in ETBF-induced inflammation. AOM-treated BALB/c mice were infected with WT-ETBF (*bft-1*) or WT-ETBF (Δ*bft-1*) and subjected to three cycles of DSS (1%) for 12 weeks. Sera were analyzed for CXCL1 and IL-17A via ELISA. Serum nitric oxide levels were examined by nitric oxide assay. (**A**) Spleen weight (mg). (**B**) Colon length (mm). (**C**) Colon weight (mg)/colon length (mm). (**D**) Serum CXCL1 levels. (**E**) Serum IL-17A levels. (**F**) Serum nitric oxide (NO) levels. Each dot represents one mouse (*n* = 6–17 mice per group). Horizontal bar, median. S, Sham; A/D, AOM/DSS alone; +E1, AOM/DSS + WT-ETBF (*bft-1*). * *p* < 0.05, ** *p* < 0.01, *** *p* < 0.001. ns, no statistical significance. Significance between treated groups was determined using the Mann–Whitney *U* test.

**Figure 5 ijms-26-06218-f005:**
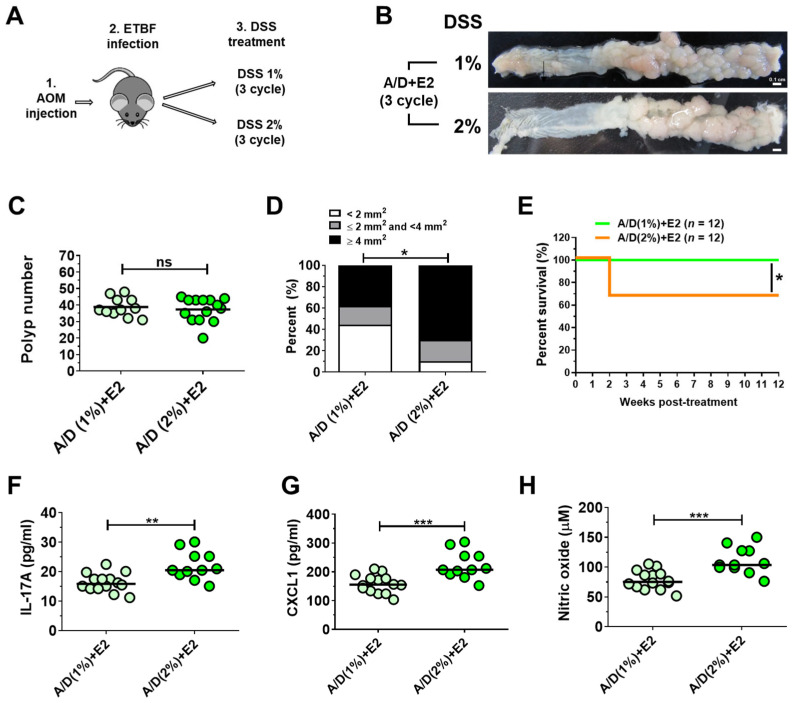
The impact of DSS dosage on the ETBF-promoted AOM/DSS model. AOM-treated BALB/c mice were infected with WT-ETBF (*bft-2*) and subjected to three cycles of DSS (1%) or DSS (2%) for 12 weeks. (**A**) Schematic illustration of experimental design. (**B**) Representative gross macroscopic image of the colon from the ETBF/AOM/DSS model according to DSS dosage (1% or 2%). Sera were analyzed for CXCL1 and IL-17A via ELISA. Serum nitric oxide levels were examined by nitric oxide assay. (**C**) Polyp number. (**D**) Polyp distribution. Distribution of polyp sizes. Statistical significance was determined using the Mann–Whitney test. ** p* < 0.05. (**E**) Survival curve of mice. Kaplan–Meier curves depicting survival of the ETBF/AOM/DSS model according to DSS dosage, Mantel–Cox log-rank test. (**F**) Serum IL-17A levels. (**G**) Serum CXCL1 levels. (**H**) Serum nitric oxide (NO) levels. Each dot represents one mouse (*n* = 11–14 mice per group). Horizontal bar, median. A/D, AOM/DSS alone; +E2, AOM/DSS + WT-ETBF (*bft-2*). * *p* < 0.05, ** *p* < 0.01, *** *p* < 0.001. ns, no statistical significance. Significance between treated groups was determined using the Mann–Whitney *U* test.

**Figure 6 ijms-26-06218-f006:**
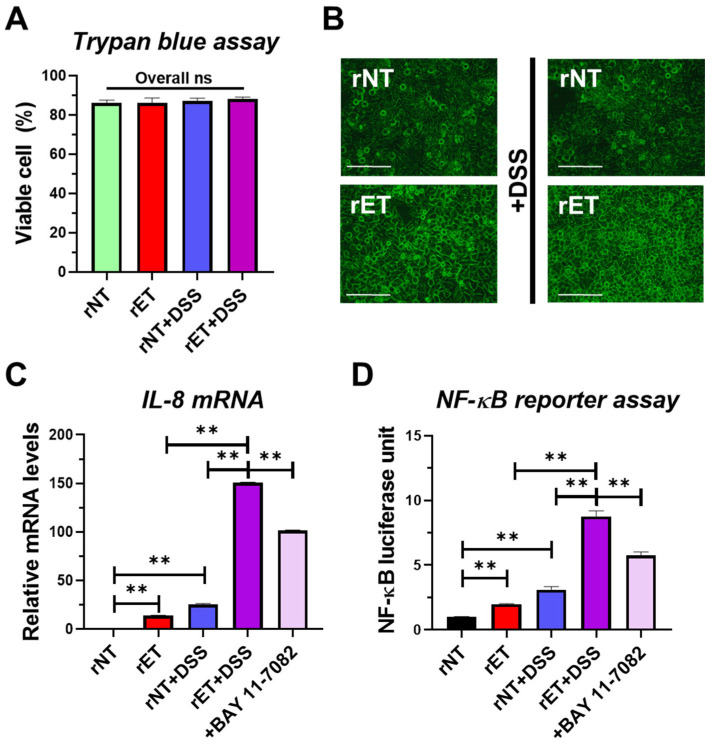
In vitro analysis of BFT and DSS effects on cell viability, IL-8 expression, and NF-κB activity in HT29/C1 cells. Human colon epithelial HT29/C1 cells were exposed to DSS either alone or in combination with rETBF (rET) or rNTBF (rNT) culture supernatants (1:10 dilution). rNT supernatant served as a negative control. (**A**) The cell viability of HT29/C1 cells treated with 1% DSS (*w*/*v*) was assessed after 24 h, both with and without rET culture supernatants (1:10). (**B**) Changes in cell morphology were observed. HT29/C1 cells were treated either with rET supernatant (positive control) alone or with rET supernatant in conjunction with 1% DSS. The morphological changes of the cells were examined using microscopy. Magnification, 400×. Scale bar, 100 μm. (**C**) qRT-PCR analysis of IL-8 expression in BFT- and/or DSS-treated HT29/C1 cells treated with Bay 11-7082 (a chemical NF-κB inhibitor; 10 μM) for 3 h. (**D**) NF-κB luciferase activity of BFT and/or DSS incubated HT29/C1 cells treated with Bay 11-7082 (10 μM) for 24 h. Luciferase activity was normalized to *Renilla* luciferase activity, and the relative values were reported. DSS, dextran sulfate sodium. Data are expressed as the mean ± SEM from three independent experiments. ** *p* < 0.01. ns, no statistical significance. Significance between treated groups was determined using the Mann–Whitney *U* test.

**Figure 7 ijms-26-06218-f007:**
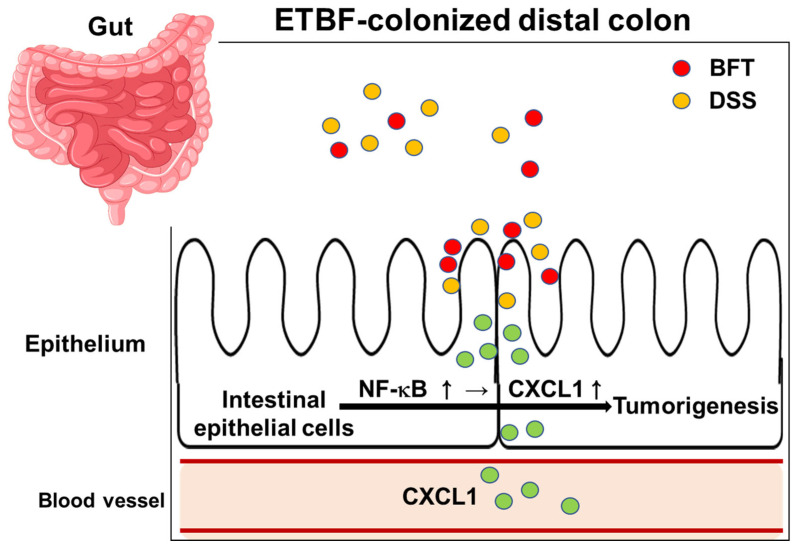
Schematic illustration of ETBF colonization-promoted AOM/DSS-mediated tumorigenesis. ETBF colonization and subsequent DSS administration synergistically promote tumorigenesis in the distal colon. Persistent BFT secretion by ETBF activates the NF-κB pathway in intestinal epithelial cells, priming them for *CXCL1* expression and secretion upon DSS exposure. This heightened inflammatory response drives tumorigenesis.

## Data Availability

The data used in this study can be obtained from the corresponding author upon reasonable request.

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
