# Peer review of "Modeling ETBF-Mediated Colorectal Tumorigenesis Using AOM/DSS in Wild-Type Mice"

_ijms, 2025, doi:10.3390/ijms26136218_

Round 1
Reviewer 1 Report
Comments and Suggestions for Authors
- What is the main question addressed by the research?
In the manuscript, Hwang et al describe the use of the well-established AOM/DSS mouse model of colorectal tumorigenesis to explore the value of some modest technical variations in the experimental protocol (mouse strains, DSS dosage) or to confirm previously reported observations (role of bft gene). They also superficially explore the link between BFT-mediated tumorigenesis and NF-κB/CXCL1 signaling in colonic epithelial cells. The manuscript is a follow up of their 2020 Int J Med Sci. paper (Ref #31; "Enterotoxigenic Bacteroides fragilis infection exacerbates tumorigenesis in AOM/DSS mouse model").
- Do you consider the topic original or relevant to the field? Does it address a specific gap in the field? Please also explain why this is/ is not the case.
- What does it add to the subject area compared with other published material?
The topic is relevant to the field. However, the new manuscript provides only a modest discovery increment in comparison to the 2020 publication, i.e., i) authors observe that BALB/c are more prone to developing colon polyps and that 1% DSS is sufficient to induce the formation of polyps upon ETBF exposure (these observations are interesting but do not address a major gap in the field); ii) authors confirmed BFT's crucial role in ETBF-promoted tumorigenesis and inflammation (this is not an original discovery, it was originally discovered using a gain of function approach in ref#31, and is now further confirmed here using a loss-of-function approach); iii) some data suggest a potential link between BFT-mediated tumorigenesis and NF-κB/CXCL1 signaling in colonic epithelial cells (the observations are correlative and do not demonstrate a functional requirement).
Accordingly, the reviewer considers that the level of originality of the manuscript is modest. The manuscript provides technical variations (mouse strains, DSS dosage) and valuable new points of reference to consider when using the ETBF/AOM/DSS model for the community of scientists who use this model of colorectal tumorigenesis. In my opinion, beyond this community, the manuscript will be of limited interest.
- What specific improvements should the authors consider regarding the methodology?
-Statistical tests used should always be named/described, e.g., Fig 1D.
-Fig. 1D, 2F: Consider showing the actual polyp size distribution as opposed to showing the % in size categories.
-Semantic: Necessity versus Sufficiency.
"Figure 3. The bft gene is solely responsible for ETBF-promoted AOM/DSS tumorigenesis."
In my humble opinion, the experiment shown in Fig 3 shows that bft is necessary. It does not explore sufficiency, which was studied in Ref #31. The statement that bft is "SOLELY" responsible is not explored in Fig. 3, only in ref #31.
On the same note, the authors state that "MPII's contribution remains less studied." It would have been interesting to test the extent to which genetic perturbation (gain or loss) of MPII has an effect, or not. Either way (hypothesis verified or not), it would have been interesting to investigate this gene but it was not tested in the manuscript.
-In Fig 6 C and D, the standard deviations shown are remarkably small. It begs the question: Do the data points represent BIOLOGICAL replicates or TECHNICAL replicates? Biological replicates should be shown, not technical replicates. For the record, the type of the replicates (biological or technical) and the number of replicates used (3, 4 or more), should be stated in the figure legend and/or methods.
-Fig 4: What does "S" stand for in the "S" control? What is the control?
-IMHO, some negative controls (no treatment or no infection) are missing in some figures. For examples: no AOM control in Fig 2; no DSS control in Fig. 5.
-Survival analyses: authors need to show the N (number of animals used) for each study group.
Are the conclusions consistent with the evidence and arguments presented and do they address the main question posed? Please also explain why this is/is not the case.
Overall, the conclusions are consistent and supported by the data shown.
For example, it is correct for the authors to note that "These results demonstrate that co-treatment with DSS and BFT elevates NF-κB/CXCL1 signaling in colonic epithelial cells, suggesting a synergistic inflammatory mechanism." In other words, the authors recognize that there is a correlation but they do not conclude (they only suggest) that there is a functionally relevant link between these two observations, which remains to be comprehensively explored.
Are the references appropriate?
Yes.
- Any additional comments on the tables and figures.
-The logic behind the use of the acronyms and the consistency of their use throughout the manuscript escape me sometimes. It could be simplified/improved for clarity.
-IMHO, Fig 7 lacks clarity, logic, consistency, and aesthetic.
-CAC acronym is not described in abstract.
Did it mean CRC?
-Fig 1A: show time scale from D1 to ~D100, not just the intervals.
-"comprehensive modeling of ETBF-associated tumorigenesis remain largely unexplored"
Not sure what this mean, exactly.
-Fig 1D legend: gray should be ">2 mm2 and < 4mm2", not "<2mm2 and <4mm2"
Author Response
Reviewer 1, General Comments 1:
In the manuscript, Hwang et al describe the use of the well-established AOM/DSS mouse model of colorectal tumorigenesis to explore the value of some modest technical variations in the experimental protocol (mouse strains, DSS dosage) or to confirm previously reported observations (role of bft gene). They also superficially explore the link between BFT-mediated tumorigenesis and NF-κB/CXCL1 signaling in colonic epithelial cells. The manuscript is a follow up of their 2020 Int J Med Sci. paper (Ref #31; "Enterotoxigenic Bacteroides fragilis infection exacerbates tumorigenesis in AOM/DSS mouse model").
Response to Reviewer 1, General Comments 1:
We agree with the reviewer’s objective summary.
Reviewer 1, General Comments 2:
The topic is relevant to the field. However, the new manuscript provides only a modest discovery increment in comparison to the 2020 publication, i.e., i) authors observe that BALB/c are more prone to developing colon polyps and that 1% DSS is sufficient to induce the formation of polyps upon ETBF exposure (these observations are interesting but do not address a major gap in the field); ii) authors confirmed BFT's crucial role in ETBF-promoted tumorigenesis and inflammation (this is not an original discovery, it was originally discovered using a gain of function approach in ref#31, and is now further confirmed here using a loss-of-function approach); iii) some data suggest a potential link between BFT-mediated tumorigenesis and NF-κB/CXCL1 signaling in colonic epithelial cells (the observations are correlative and do not demonstrate a functional requirement). Accordingly, the reviewer considers that the level of originality of the manuscript is modest. The manuscript provides technical variations (mouse strains, DSS dosage) and valuable new points of reference to consider when using the ETBF/AOM/DSS model for the community of scientists who use this model of colorectal tumorigenesis. In my opinion, beyond this community, the manuscript will be of limited interest.
Response to Reviewer 1, General Comments 2:
We agree with the reviewer’s concise summary of our manuscript.
Reviewer 1, Specific Comment 1:
Statistical tests used should always be named/described, e.g., Fig 1D.
Response to Reviewer 1, Specific Comment 1:
We sincerely appreciate the constructive feedback. In response, we have included the specific statistical tests used for Figures 1D, 2D, and 5D in the respective figure legends. We believe that this addition has enhanced the clarity and overall quality of the manuscript.
Reviewer 1, Specific Comment 2:
Fig. 1D, 2F: Consider showing the actual polyp size distribution as opposed to showing the % in size categories.
Response to Reviewer 1, Specific Comment 2:
We appreciate the reviewer’s constructive suggestion. During the initial stages of manuscript preparation, we also considered presenting the actual polyp size distribution. However, due to the large number of polyps generated in our model, we found that plotting individual data points (e.g., dot or line plots) would make it difficult to intuitively convey the extent of tumor burden. As such, we opted to present the data using percentage-based size categories, which we believe more effectively highlight the overall trends in polyp growth. We hope this rationale sufficiently addresses the reviewer’s concern.
Reviewer 1, Specific Comment 3:
Semantic: Necessity versus Sufficiency. "Figure 3. The bft gene is solely responsible for ETBF-promoted AOM/DSS tumorigenesis." In my humble opinion, the experiment shown in Fig 3 shows that bft is necessary. It does not explore sufficiency, which was studied in Ref #31. The statement that bft is "SOLELY" responsible is not explored in Fig. 3, only in ref #31.
Response to Reviewer 1, Specific Comment 3:
We appreciate and agree with the reviewer’s insightful comment. In response, we have revised the title of Figure 3 as follows to more accurately reflect the data presented. The revised title now reads as follows, “Figure 3. The bft gene is necessary for ETBF-promoted AOM/DSS tumorigenesis.”
Reviewer 1, Specific Comment 4:
On the same note, the authors state that "MPII's contribution remains less studied." It would have been interesting to test the extent to which genetic perturbation (gain or loss) of MPII has an effect, or not. Either way (hypothesis verified or not), it would have been interesting to investigate this gene but it was not tested in the manuscript.
Response to Reviewer 1, Specific Comment 4:
We sincerely appreciate the reviewer’s insightful suggestion. As correctly noted, our current study does not explore the function of the mp2 gene using a gain-of-function approach. We agree that this remains an important area of investigation. As a follow-up to this study, we plan to construct a recombinant Bacteroides fragilis strain expressing mp2 via a plasmid-based system. This approach will allow us to evaluate the direct contribution of MP2 in ETBF-associated tumorigenesis in greater detail. We greatly appreciate the reviewer’s valuable recommendation, which will help guide the direction of our future work.
Reviewer 1, Specific Comment 5:
In Fig 6 C and D, the standard deviations shown are remarkably small. It begs the question: Do the data points represent BIOLOGICAL replicates or TECHNICAL replicates? Biological replicates should be shown, not technical replicates. For the record, the type of the replicates (biological or technical) and the number of replicates used (3, 4 or more), should be stated in the figure legend and/or methods.
Response to Reviewer 1, Specific Comment 5:
We thank the reviewer for the important comment. The data shown in Figures 6C and 6D represent biological replicates, not technical replicates. Specifically, the experiments were independently performed three times, and results are presented as mean ± SEM. We have clarified this in the figure legends and ensured that the number and type of replicates are now explicitly stated in both the Methods section and figure captions.
Reviewer 1, Specific Comment 6:
Fig 4: What does "S" stand for in the "S" control? What is the control?
Response to Reviewer 1, Specific Comment 6:
The abbreviation “S” refers to Sham, which denotes a mock or placebo-treated group, and functionally serves as a control in this study. We have revised the legends for Figures 3 and 4 to include this definition for clarity.
Reviewer 1, Specific Comment 7:
Some negative controls (no treatment or no infection) are missing in some figures. For examples: no AOM control in Fig 2; no DSS control in Fig. 5.
Response to Reviewer 1, Specific Comment 7:
As reported in our group’s previous publication in the International Journal of Medical Sciences (2020), the AOM-only and DSS-only control groups were included and thoroughly analyzed. It was clearly demonstrated that neither AOM nor DSS treatment alone induces polyp formation. Therefore, we consider these conditions to be biologically irrelevant controls when assessing the efficiency of polyp generation. We hope this rationale is sufficiently convincing and helps clarify our decision to exclude these groups from the current experimental design.
Reviewer 1, Specific Comment 8:
Survival analyses: authors need to show the N (number of animals used) for each study group.
Response to Reviewer 1, Specific Comment 8:
We sincerely appreciate the reviewer’s constructive feedback. In response, we have indicated the number of animals used per group (n) directly in the survival curves presented in Figures 3 and 5. We hope this revision reflects our commitment to enhancing the clarity and overall quality of the manuscript.
Reviewer 1, General Comments 3:
Overall, the conclusions are consistent and supported by the data shown.
Response to Reviewer 1, General Comments 3:
We thank the reviewer for the supportive comment. We are glad that the conclusions were found to be consistent with the data presented. We appreciate your thoughtful evaluation.
Reviewer 1, General Comments 4:
For example, it is correct for the authors to note that "These results demonstrate that co-treatment with DSS and BFT elevates NF-κB/CXCL1 signaling in colonic epithelial cells, suggesting a synergistic inflammatory mechanism." In other words, the authors recognize that there is a correlation but they do not conclude (they only suggest) that there is a functionally relevant link between these two observations, which remains to be comprehensively explored.
Response to Reviewer 1, General Comments 4:
We appreciate the reviewer’s thoughtful assessment. Indeed, our intention was to highlight a potential synergistic inflammatory response without overstating causality. We fully agree that the functional significance of the observed NF-κB/CXCL1 activation remains to be elucidated through further mechanistic studies, which are currently underway in our laboratory.
Reviewer 1, Specific Comment 9:
The logic behind the use of the acronyms and the consistency of their use throughout the manuscript escape me sometimes. It could be simplified/improved for clarity.
Response to Reviewer 1, Specific Comment 9:
We thank the reviewer for pointing this out. We have carefully reviewed the use of acronyms throughout the manuscript and revised several instances to improve clarity and consistency. Specifically, we ensured that each acronym is clearly defined upon first use and applied uniformly across the main text, tables, and figure legends. Additionally, in response to a related comment from another reviewer, we have corrected the inconsistent use of the terms, tumor and polyp. These terms have now been standardized to maintain terminological consistency throughout the manuscript, in line with the biological context of the study. We sincerely appreciate the reviewer’s helpful feedback, which has contributed to enhancing the clarity and readability of the manuscript.
Reviewer 1, Specific Comment 10:
Fig 7 lacks clarity, logic, consistency, and aesthetic.
Response to Reviewer 1, Specific Comment 10:
We regret that the intended meaning of Figure 7 may not have been clearly conveyed. However, we would like to note that the rationale and interpretation behind this figure are described in detail in the accompanying figure legend. We kindly invite the reviewer to refer to this explanation, along with our revised responses, in the hope that our intended message is now more effectively communicated through the current version of the manuscript.
Reviewer 1, Specific Comment 11:
CAC acronym is not described in abstract. Did it mean CRC?
Response to Reviewer 1, Specific Comment 11:
We would like to clarify that “CAC” refers to colitis-associated cancer. In response to this point, we have revised the manuscript to replace all instances of the abbreviation “CAC” with the full term colitis-associated cancer to enhance clarity and consistency throughout the text.
Reviewer 1, Specific Comment 12:
Fig 1A: show time scale from D1 to ~D100, not just the intervals.
Response to Reviewer 1, Specific Comment 12:
We thank the reviewer for the constructive feedback. In response, we have added specific time points (D1 to D84) to Figure 1A to clearly indicate the experimental timeline. We hope this revision improves the overall readability and clarity of the experimental design.
Reviewer 1, Specific Comment 13:
"comprehensive modeling of ETBF-associated tumorigenesis remain largely unexplored". Not sure what this mean, exactly.
Response to Reviewer 1, Specific Comment 13:
We thank the reviewer for the thoughtful question. The original statement was intended to convey that the lack of well-defined in vivo systems to comprehensively investigate the mechanisms by which ETBF promotes colitis-associated cancer remains an important unresolved issue. We have revised and clarified the sentence accordingly in the manuscript, and we hope the updated version more accurately reflects our intended meaning.
Reviewer 1, Specific Comment 14:
Fig 1D legend: gray should be ">2 mm2 and < 4mm2", not "<2mm2 and <4mm2"
Response to Reviewer 1, Specific Comment 14:
We sincerely thank the reviewer for the attentive comment. As suggested, we have revised and refined the legend for Figure 1D accordingly.

Reviewer 2 Report
Comments and Suggestions for Authors
Dear authors,
I have thoroughly gone through your Manuscript Draft entitled “Modeling ETBF-Mediated Colorectal Tumorigenesis Using AOM/DSS in Wild-Type Mice” This is an excellent manuscript having deep insight into every aspect of the study. The following minor changes can make this script easier and more understandable for readers.
I am mentioning here with section and line no, please make corrections accordingly.
- Add few more keywords that will diversify search spectrum of the manuscript, for easy access of reader.
- Introduction Page No 1 Line no 2 add latest epidemiological statistics, and reference accordingly.
- Introduction Page No 2 2nd paragraph Line no 8, 9 add a few lines explaining CYP2E1-medi ated metabolism.
- Introduction Page No 2 2nd paragraph Line no 16 add a few words explaining about dosage of AOM injections.
- The author should discuss the overall importance of gut microbiota, the gut microbiota is crucial for host health, regulating digestion, nutrient absorption, immune function, and protection against pathogens. Dysbiosis can lead to diseases like IBD, CRC, metabolic disorders, and neurodegenerative conditions. Importantly, the gut microbiota also influences the outcomes of experimental models of disease, such as the AOM/DSS-induced colitis model, where microbial composition can affect inflammation severity and tumor development. Emphasizing the overall importance of gut microbiota provides a broader biological context and highlights why microbiota-related factors must be considered in both mechanistic studies and therapeutic research.
- DSS (Dextran Sulfate Sodium) is a standard model for studying intestinal inflammation and colitis-associated cancer (especially when combined with ETBF/AOM/DSS). However, DSS can cause significant weight loss, dehydration, severe colitis, and even death the, Author haven’t mentioned about mortality rate, animal groups etc.
Author Response
Reviewer 2, Comment 1:
Dear authors, I have thoroughly gone through your Manuscript Draft entitled “Modeling ETBF-Mediated Colorectal Tumorigenesis Using AOM/DSS in Wild-Type Mice” This is an excellent manuscript having deep insight into every aspect of the study. The following minor changes can make this script easier and more understandable for readers. I am mentioning here with section and line no, please make corrections accordingly.
Response to Reviewer 2, Comment 1:
Thank you for your positive overall assessment. We will carefully address the points raised below to enhance the quality of the manuscript.
Reviewer 2, Comment 2:
Add few more keywords that will diversify search spectrum of the manuscript, for easy access of reader.
Response to Reviewer 2, Comment 2:
We appreciate your constructive feedback. To enhance broader accessibility for readers, we have incorporated two additional keywords (Colitis and DSS) into the abstract.
Reviewer 2, Comment 3:
Introduction Page No 1 Line no 2 add latest epidemiological statistics, and reference accordingly.
Response to Reviewer 2, Comment 3:
As per your suggestion, two newly selected references on recent epidemiological statistics have been added to the previously included literature, further strengthening the study’s foundation. The following references have been newly incorporated into the study:
- Lavelle et al. (2022) investigated fecal microbiota and bile acids in inflammatory bowel disease (IBD) patients undergoing colorectal cancer screening (Gut Microbes, 14(1), 2078620).
- Uchino et al. (2024) conducted a nationwide propensity-score-matched analysis to differentiate histological characteristics between sporadic and colitis-associated intestinal cancers (J Gastroenterol Hepatol, 39(5), 893–901)
Reviewer 2, Comment 4:
Introduction Page No 2 2nd paragraph Line no 8, 9 add a few lines explaining CYP2E1-mediated metabolism.
Response to Reviewer 2, Comment 4:
Thank you for identifying the areas that required further clarification. In response, we have provided a more detailed explanation of the CYP2E1-mediated metabolic activation of AOM below (see lines 61-63). The following revised content has been newly incorporated. “Azoxymethane (AOM) undergoes metabolic activation via cytochrome P450 2E1 (CYP2E1), which catalyzes hydroxylation at the methyl group distal to the N(O) functional moiety, converting AOM into methylazoxymethanol (MAM).”
Reviewer 2, Comment 5:
Introduction Page No 2 2nd paragraph Line no 16 add a few words explaining about dosage of AOM injections.
Response to Reviewer 2, Comment 5:
As per your suggestion, both the AOM dosage and DSS treatment concentrations have been revised and explicitly specified. The following revised content has been newly incorporated, “However, the dosages of AOM (5–12.5 mg/kg) and DSS (1–3%) vary considerably across studies.”
Reviewer 2, Comment 6:
The author should discuss the overall importance of gut microbiota, the gut microbiota is crucial for host health, regulating digestion, nutrient absorption, immune function, and protection against pathogens. Dysbiosis can lead to diseases like IBD, CRC, metabolic disorders, and neurodegenerative conditions. Importantly, the gut microbiota also influences the outcomes of experimental models of disease, such as the AOM/DSS-induced colitis model, where microbial composition can affect inflammation severity and tumor development. Emphasizing the overall importance of gut microbiota provides a broader biological context and highlights why microbiota-related factors must be considered in both mechanistic studies and therapeutic research.
Response to Reviewer 2, Comment 6:
We acknowledge the reviewer’s comments and fully agree with your recommendations. Accordingly, the discussion has been revised to incorporate the significance of gut microbiota in human health, as well as its role in ETBF-mediated colorectal polyp formation. This highlights the necessity for further research on microbiota-related mechanisms involved in tumorigenesis. These additions have strengthened the quality of the manuscript. Thank you for your valuable insights. The following revised content has been newly incorporated, “The gut microbiota plays a crucial role in maintaining host health by regulating digestion, nutrient absorption, immune responses, and defense against pathogens [60]. In experimental models such as AOM/DSS-induced colitis, microbial composition influences the severity of inflammation and tumor development [61, 62], emphasizing its significance in disease progression. Given this impact, gut microbiota should be a key consideration in mechanistic studies on carcinogenesis, particularly in ETBF-mediated colorectal polyp formation, where microbial interactions may shape disease outcomes. Exploring the microbiome’s role in ETBF-induced tumorigenesis could deepen our understanding of microbial contributions to cancer progression and support the development of microbiota-targeted therapeutic approaches. Advancing knowledge in this field will strengthen colorectal cancer research, highlighting the microbiome’s relevance in both experimental and clinical studies.”
Reviewer 2, Comment 7:
DSS (Dextran Sulfate Sodium) is a standard model for studying intestinal inflammation and colitis-associated cancer (especially when combined with ETBF/AOM/DSS). However, DSS can cause significant weight loss, dehydration, severe colitis, and even death the, Author haven’t mentioned about mortality rate, animal groups etc.
Response to Reviewer 2, Comment 7:
We sincerely appreciate the attention to detail by the reviewer. This study applies the ETBF-induced colorectal polyp model within the AOM/DSS system, with survival data presented in graphical form in Figure 3D and Figure 5E. As noted, specific survival percentages were not explicitly stated in the Results section. To address this, precise survival rates for each experimental group have been incorporated into the Figure 3 (see lines 173-176) and Figure 5 results (see lines 224-228), ensuring a more comprehensive representation of the data.
The paragraph describing the Figure 3 results in the revised manuscript reads as follows:
“The survival rate in the bft-1 deletion ETBF-infected group remained unchanged at 100%, comparable to the AOM/DSS control group. In contrast, the WT-ETBF-infected group exhibited a significantly lower survival rate (63.6%) (Figure 3D).”
The paragraph describing the Figure 5 results in the revised manuscript reads as follows:
“However, the average tumor volume in the ETBF/AOM/DSS (2%) group was significantly greater than that in the 1% DSS group (Figure 5D). Conversely, the survival rate in the 2% DSS group was markedly lower (66.7%) compared to the 1% DSS group, which exhibited full survival (100%) (Figure 5E).”

Reviewer 3 Report
Comments and Suggestions for Authors
The design concept of this paper is very good and had conducted sufficient research content, but further improvement is needed as listed below:
Page 7: The DSS dosing concentration set by the experimental group is too simple, it is recommended to increase it.
Page 8 (Figure 6): Does the experimental group have separate results for DSS? Based on the current results, the “rET+DSS” experimental group performed much better than “rET”, and a reasonable validation or explanation is needed for this result!
Page 9: “Initially, we performed a trypan blue assay to confirm that the combination of DSS and rET (the culture supernatant of recombinant B. fragilis secreting active BFT) did not directly decrease cell viability (Figure 6A), thus ruling out a cytotoxic effect as the cause of increased inflammation.” Does this conclusion contradict the content of the introduction?
Disscussion: Delete some repetitive descriptions in the discussion and introduction. The discussion section needs to be further strengthened! Especially for the main results, a more detailed discussion is needed to highlight the highlights and value of this research.
Author Response
Reviewer 3, Comment 1:
The design concept of this paper is very good and had conducted sufficient research content, but further improvement is needed as listed below:
Response to Reviewer 3, Comment 1:
Thank you for your positive comments. We will carefully reflect on the points you have mentioned and improve the manuscript.
Reviewer 3, Comment 2:
Page 7: The DSS dosing concentration set by the experimental group is too simple, it is recommended to increase it.
Response to Reviewer 3, Comment 2:
Our research group partially agrees with the concerns raised. However, the concentration of DSS (Dextran Sulfate Sodium) used in the traditional AOM/DSS model has been largely standardized within a typical range of 1–2.5%. Accordingly, we conducted our experiments and structured our research paper using the most commonly utilized DSS concentrations, specifically 1% and 2%. In our optimized ETBF polyp model, the use of only 1% DSS has proven to be sufficient for effective polyp formation, presenting a significant economic advantage. Additionally, employing a relatively lower DSS concentration reduces the distress experienced by the mice, thus minimizing their suffering. We hope this response sufficiently addresses the concerns raised.
Reviewer 3, Comment 3:
Page 8 (Figure 6): Does the experimental group have separate results for DSS? Based on the current results, the “rET+DSS” experimental group performed much better than “rET”, and a reasonable validation or explanation is needed for this result!
Response to Reviewer 3, Comment 3:
There is no experimental group treated solely with DSS in the current submitted manuscript. However, the rNT+DSS group can serve as a comparable substitute representing the DSS-only treatment group. The supernatant obtained from the cultured rNTBF strain was filtered to produce media, which, when applied to intestinal epithelial cells, contains inactive BFT. Since inactive BFT does not induce E-cadherin cleavage, it does not trigger NF-kB/IL-8 expression. Therefore, the most appropriate control group for the rET-only group, which contains active BFT, is the rNT group, which includes inactive BFT. We hope this response sufficiently addresses the concerns raised.
Reviewer 3, Comment 4:
Page 9: “Initially, we performed a trypan blue assay to confirm that the combination of DSS and rET (the culture supernatant of recombinant B. fragilis secreting active BFT) did not directly decrease cell viability (Figure 6A), thus ruling out a cytotoxic effect as the cause of increased inflammation.” Does this conclusion contradict the content of the introduction?
Response to Reviewer 3, Comment 4:
Pathologically, "inflammation" refers to a physiological response that involves eliminating invading microorganisms and removing damaged host cells. While DSS and BFT exert distinct mechanisms, both can individually induce damage to intestinal epithelial cells. However, in the HT29/C1 cell line used in our study, simultaneous treatment with BFT and DSS did not result in cell death or overt cytotoxic effects. Thus, we concluded that the increase in in vivo inflammation was not due to epithelial cell death. This conclusion is described in Figure 6A and does not contradict the introduction’s discussion of DSS-induced inflammation; rather, it conveys a different implication. We hope this explanation sufficiently clarifies the matter.
Reviewer 3, Comment 5:
Disscussion: Delete some repetitive descriptions in the discussion and introduction. The discussion section needs to be further strengthened! Especially for the main results, a more detailed discussion is needed to highlight the highlights and value of this research.
Response to Reviewer 3, Comment 5:
To enhance clarity and avoid redundancy, we have refined the discussion section by condensing paragraphs that significantly overlapped with the introduction. These revisions ensure a more concise presentation of key findings while minimizing repetition (see line 299-306). Additionally, while this study extensively investigates colorectal cancer models, it does not fully address the critical role of gut microbiota in CRC development. Given the dynamic changes in pathogenic microbiota within the ETBF-induced colorectal polyp model, further research is warranted to explore how these microbial alterations contribute to tumorigenesis. A new section has been incorporated into the discussion to emphasize the need for studies examining the functional impact of these microbiota shifts in ETBF-driven colonic pathology (see line 363-374). We hope that our efforts to improve the manuscript are well conveyed and contribute meaningfully to its overall quality.

Reviewer 4 Report
Comments and Suggestions for Authors
The authors here present an interesting manuscript on the impact of ETBF on tumors within the colon. However I found many components of the manuscript, although fairly well written, to be very confusing. I suggest major revisions. See my comments below.
Abstract.
The authors do not define many acronyms in the abstract including BALB/C. NF-xB etc. Also the authors assume the reader understands AOM/DSS after not defining or articulating this here or in the intro. Same with what these drugs cover/how they're used. Elaboration is needed.
Intro.
As someone who does not work with mice is needs to be clarified why these models are used and how they differ. it is not clear in the text.
Figures.
The figures are unclear- for example they put 1A as a timeline but right underneath is a picture of the intestine. It makes the readers think it is associated with the timeline itself. And please place arrows on ALL tumor growths in ALL pictures. If the reader is to believe this is bein properly quanitified, we should be able to count them for ourselves.
Figure is 7 is VERY unclear. No idea what this is trying to do.
Methods.
Although I do not work with mice and can't really speak to the experiments used here (they seem fine but maybe another reviewer would know) I will say that it is very important to document exactly HOW MANY mice are being used in each group. This is not articulated in the text and very important to the results. Please include a table. The results get a bit confusing with all of the metrics so it would be easier to lay out all the work that is done like that.
Discussion.
Please incoporate potential clinical applicaitons of this. If it has been used in a human clinical setting please cite. Please also clarify more about the mouse strains here.
Data availability.
Please place all pictures and data in the supplemental files. It is not appropriate to ask readers to email the authors.
Author Response
Reviewer 4, Comment 1:
The authors here present an interesting manuscript on the impact of ETBF on tumors within the colon. However, I found many components of the manuscript, although fairly well written, to be very confusing. I suggest major revisions. See my comments below.
Response to Reviewer 4, Comment 1:
We have tried to respond to each critique as well as possible and incorporate the feedback provided.
Reviewer 4, Comment 2:
Abstract. The authors do not define many acronyms in the abstract including BALB/C. NF-xB etc.
Response to Reviewer 4, Comment 2:
Thank you for your feedback. We appreciate the careful review of our abstract. Upon reassessment, we found that NF-κB was already included in the abbreviation list. We acknowledge that the full name for BALB/c was missing, and we have now added it to the abbreviation section. Without offending the reviewer, generally the BALB/c strain is normally not written in full form and used as written. Thanks to the reviewer, I have learned the abbreviation of “BALB/c” while writing the response letter! Additionally, we revisited the introduction to enhance explanations regarding AOM and DSS, including their applications and mechanisms, to improve readability and comprehension. We value your suggestions and made the necessary refinements accordingly.
Reviewer 4, Comment 3:
Also, the authors assume the reader understands AOM/DSS after not defining or articulating this here or in the intro. Same with what these drugs cover/how they're used. Elaboration is needed.
Response to Reviewer 4, Comment 3:
The AOM/DSS system is one of the most commonly utilized methods in colorectal cancer research. Given its technical complexity, providing a detailed explanation in the introduction may not be suitable. Instead, we have included a comprehensive description of its application in the Methods section. Specifically, the Mouse Experiments subsection elaborates on the usage of AOM and DSS. We hope that this revision adds clarity to the reviewer’s concerns.
Reviewer 4, Comment 4:
Intro. As someone who does not work with mice is needs to be clarified why these models are used and how they differ. it is not clear in the text.
Response to Reviewer 4, Comment 4:
To investigate the carcinogenesis process of enterotoxigenic Bacteroides fragilis (ETBF)-mediated colorectal cancer in humans at an experimental level and study its underlying mechanisms, animal models are essential. The conventional azoxymethane/dextran sulfate sodium (AOM/DSS) polyp model does not replicate an intestinal microenvironment colonized by ETBF, necessitating a murine model where colorectal cancer is promoted by ETBF. Such a model is critical for elucidating the molecular and cellular biological mechanisms underlying ETBF-induced carcinogenesis, as well as for understanding the phenotypic characteristics of colorectal cancer observed in humans. Given these considerations, the ETBF-associated colorectal polyp model holds significant scientific value. The purpose and significance of this study is outlined in the final paragraph of the Discussion section. We hope the response was sufficiently satisfactory.
Reviewer 4, Comment 5:
Figures. The figures are unclear- for example they put 1A as a timeline but right underneath is a picture of the intestine. It makes the readers think it is associated with the timeline itself.
Response to Reviewer 4, Comment 5:
We appreciate the reviewer’s comments regarding the clarity of the figures and the presentation of tumor quantification. Regarding Figure 1A, our intention was to provide a visual timeline of the experimental design while also presenting relevant histological images.
Reviewer 4, Comment 6:
And please place arrows on ALL tumor growths in ALL pictures. If the reader is to believe this is being properly quantified, we should be able to count them for ourselves.
Response to Reviewer 4, Comment 6:
In Figure 1, considering that some researchers may be unfamiliar with identifying polyps, we have ensured that all visible polyps are clearly marked with arrows to facilitate interpretation. In the subsequent figures, however, we have intentionally omitted additional arrow markers to prioritize image readability and maintain clarity. We hope that these efforts adequately address the concern and contribute to a more effective presentation of the data.
Reviewer 4, Comment 7:
Figure is 7 is VERY unclear. No idea what this is trying to do.
Response to Reviewer 4, Comment 7:
Figure 7 was designed to visually represent the progression of colonic mucosal inflammation exacerbated by enterotoxigenic Bacteroides fragilis (ETBF) and dextran sulfate sodium (DSS), as well as the subsequent maturation of colorectal polyps. Specifically, this figure illustrates how ETBF-associated bacterial activity drives rapid polyp formation and inflammatory responses within the AOM/DSS model, highlighting that these pathological effects are mediated by Bacteroides fragilis toxin (BFT) secretion and DSS exposure. Given its intent and scientific relevance, we hope that this summary figure will be recognized for effectively conveying the core findings of this study.
Reviewer 4, Comment 8:
Methods. Although I do not work with mice and can't really speak to the experiments used here (they seem fine but maybe another reviewer would know) I will say that it is very important to document exactly HOW MANY mice are being used in each group. This is not articulated in the text and very important to the results. Please include a table. The results get a bit confusing with all of the metrics so it would be easier to lay out all the work that is done like that.
Response to Reviewer 4, Comment 8:
The figure legend provides the minimum and maximum number (i.e., range) of mice used per group in each figure. Because the exact number differs due to various circumstances (eg, death). We can only respond that the range of mice numbers are generally provided in our field of study. Additionally, while a table specifying the exact number of mice per figure is not included, the dot plots within the figures allows a visual representation of the mice used per group. We hope this explanation effectively conveys that the graphical presentation in our study is both logically structured and intuitively designed for clear interpretation.
Reviewer 4, Comment 9:
Discussion. Please incorporate potential clinical applications of this. If it has been used in a human clinical setting please cite. Please also clarify more about the mouse strains here.
Response to Reviewer 4, Comment 9:
In lines 43-44 of the Introduction, the high prevalence of enterotoxigenic Bacteroides fragilis (ETBF) colonization in colorectal cancer patients has been discussed, with relevant clinical studies properly cited to support this claim. Additionally, information on colon polyp susceptibility across different mouse strains has been incorporated into the Discussion section. Specifically, we compare BALB/c and C57BL/6 mice, addressing their respective responses to polyp formation while also integrating findings from other research groups to provide a broader perspective. We hope that these additions sufficiently enhance the discussion on mouse strain variability, ensuring a comprehensive and well-supported analysis.
Reviewer 4, Comment 10:
Data availability. Please place all pictures and data in the supplemental files. It is not appropriate to ask readers to email the authors.
Response to Reviewer 4, Comment 10:
As per the journal’s guidelines, we have followed the standard data-sharing policies by indicating that additional materials—including images and datasets—can be obtained upon request from the corresponding authors. This approach aligns with established publication practices, ensuring appropriate access to supplementary data while maintaining ethical considerations regarding research transparency.

Reviewer 5 Report
Comments and Suggestions for Authors
This study aims to refine colitis associated colorectal cancer AOM/DSS model by integrating enterotoxigenic Bacterioides fragilis (ETBF) infection.
Key points:
- single low-dose (1 %) DSS cycle is sufficient to elicit robust tumour formation with minimal mortality
- BFT is the sole ETBF virulence factor that synergises with DSS via NF-κB/CXCL1 signalling to drive polyp growth.
Strengths lie in the systematic optimization of model variables (mouse strain, DSS dose), genetic proof of BFT dependence, and in the proposal of a new ETBF/AOM/DSS model for studying the role of microbiota in CRC carcinogenesis and potential therapeutic interventions. However, since similar ETBF-AOM/DSS work has been published previously (Hwang et al., 2020, 2024), its advance is incremental.
Materials and methods are described in sufficient detail and correspond with the aim of the study. Further points which require explanations are the following:
- Why only female mice were used.
- Randomisation for polyp counting is not stated.
Discussion section would benefit from the paragraph on the translatability to human CRC.
Further points to explain and discuss:
- Resident microbiota is seriously affected by the clindamycin + gentamicin, and this can attenuate or exacerbate AOM/DSS tumor burden and cytokine levels (PMID: 31860645). It therefore becomes difficult to explain how much of the tumor promoting effect arises from loss of protective commensals, antibiotic-induced epithelial stress or BFT signalling.
Therefore it is necessary to explain why high-dose antibiotics are essential here and discuss their potential tumor promoting or tumor inhibitory effects.
- Further optimizations which would strengthen the translational relevance would include a no-antibiotic ETBF group which would demonstrate that BFT-dependent tumor promotion persists when the background microbiota is intact.
The terms tumor and polyp should be used consistently through the text, there is mention of tumor incidence in the text (eg. when referring to Figure 5B, C), while the Figure 5 text description mentions polyp number and polyp distribution. So either a colonic histopathology (representative H&E sections) or a literature reference explaining this is advised.
Author Response
Reviewer 5, Comment 1:
This study aims to refine colitis associated colorectal cancer AOM/DSS model by integrating enterotoxigenic Bacterioides fragilis (ETBF) infection.
Key points:
- single low-dose (1 %) DSS cycle is sufficient to elicit robust tumour formation with minimal mortality
- BFT is the sole ETBF virulence factor that synergises with DSS via NF-κB/CXCL1 signalling to drive polyp growth.
Strengths lie in the systematic optimization of model variables (mouse strain, DSS dose), genetic proof of BFT dependence, and in the proposal of a new ETBF/AOM/DSS model for studying the role of microbiota in CRC carcinogenesis and potential therapeutic interventions. However, since similar ETBF-AOM/DSS work has been published previously (Hwang et al., 2020, 2024), its advance is incremental.
Response to Reviewer 5, Comment 1:
Thank you for acknowledging the key contributions and strengths of our manuscript. While we recognize some of the concerns raised regarding its limitations, we would like to emphasize that the special issue to which we submitted our paper primarily focuses on the technical aspects of establishing a colorectal polyp induction model for studying gut microbiota. It also provides detailed information on the dosage and duration of the compounds capable of inducing polyps. Additionally, we highlight the finding that simultaneous exposure of intestinal epithelial cells to DSS and BFT enterotoxin promote cancer progression through the activation of inflammatory pathways rather than inducing cell death.
Reviewer 5, Comment 2:
Materials and methods are described in sufficient detail and correspond with the aim of the study. Further points which require explanations are the following: Why only female mice were used.
Response to Reviewer 5, Comment 2:
Our research group specifically selected female mice for the long-term colorectal cancer induction protocol due to their reduced inter-individual aggression compared to males, which made them more manageable in experimental settings. According to the study by Chung et al. (2018) published in Cell Host & Microbe, there is no significant difference in the severity of colonic inflammation induced by enterotoxigenic Bacteroides fragilis (ETBF) between male and female mice. Therefore, all further experiments were thus conducted using female mice.
Reviewer 5, Comment 3:
Materials and methods are described in sufficient detail and correspond with the aim of the study. Further points which require explanations are the following: Randomisation for polyp counting is not stated.
Response to Reviewer 5, Comment 3:
We did not perform randomization of mice for polyp enumeration. We tried our best to conduct a non-biased evaluation of polyp number and size, but the evaluator was aware of the group designation. In hindsight, evaluation of polyps in a “blinded” manner will have been more desired. We can only state our methods as truthfully as possible. We thank the reviewer for this comment and we will incorporate this in all future studies.
Reviewer 5, Comment 4:
Discussion section would benefit from the paragraph on the translatability to human CRC.
Response to Reviewer 5, Comment 4:
In the Discussion section, we have supplemented the existing review literature by incorporating a discussion on the susceptibility of the ETBF polyp model across different mouse strains (see lines 304–320). Additionally, we have expanded our analysis to examine the role of other gut microbiota in polyp development induced by ETBF (see lines 385–396). We hope that these additions sufficiently address the need for further discussion and enhance the overall depth of our manuscript.
Reviewer 5, Comment 5:
Resident microbiota is seriously affected by the clindamycin + gentamicin, and this can attenuate or exacerbate AOM/DSS tumor burden and cytokine levels (PMID: 31860645). It therefore becomes difficult to explain how much of the tumor promoting effect arises from loss of protective commensals, antibiotic-induced epithelial stress or BFT signaling. Therefore, it is necessary to explain why high-dose antibiotics are essential here and discuss their potential tumor promoting or tumor inhibitory effects. Further optimizations which would strengthen the translational relevance would include a no-antibiotic ETBF group which would demonstrate that BFT-dependent tumor promotion persists when the background microbiota is intact.
Response to Reviewer 5, Comment 5:
In the experimental protocol used in our study, antibiotics were initially administered to facilitate the colonization of ETBF in the mouse intestine. We found early on in our studies that, in the absence of antibiotic treatment, bacterial colonization was inconsistent between the mice within the same groups. Therefore, the main reason for the initial antibiotic treatment was the synchronize uniform bacterial colonization. However, no additional antibiotic treatment was given during the DSS cycles. While the early depletion of commensal microbiota may have contributed to tumor-promoting effects, this loss occurred uniformly across both the experimental and control groups due to antibiotic treatment. Given these conditions, our findings suggest that the observed increase in colonic polyps was primarily driven by the pathogenic factors associated with BFT toxin produced by ETBF. Interestingly, the study by Zackular et al. (mBio, 2013) reported that antibiotic treatment suppressed colonic polyp formation induced by AOM/DSS. This contrast highlights the complexity of host-microbiota interactions in colorectal tumorigenesis and underscores the need for further investigation into the mechanisms underlying these divergent effects.
Reviewer 5, Comment 6:
The terms tumor and polyp should be used consistently through the text, there is mention of tumor incidence in the text (eg. when referring to Figure 5B, C), while the Figure 5 text description mentions polyp number and polyp distribution. So, either a colonic histopathology (representative H&E sections) or a literature reference explaining this is advised.
Response to Reviewer 5, Comment 6:
We have revised the terminology in the Figure 5 text, replacing "tumor" with "polyp" to ensure consistency throughout the manuscript. This adjustment has helped maintain uniformity in terminology across the text, enhancing clarity and coherence. Thank you for your valuable feedback.

Round 2
Reviewer 1 Report
Comments and Suggestions for Authors
I have no further comments.
Author Response
We hope that our responses provided in the first round of revision were clearly conveyed to the reviewer. Moving forward, we will carefully incorporate the comments from the other reviewers to further enhance the quality of the manuscript.
We sincerely thank you for taking the time to review our research work.
Reviewer 3 Report
Comments and Suggestions for Authors
After the manuscript was revised, it basically met the standards for publication in the journal, but there are still a few small areas that need further improvement:
- There are still a few writing errors in the text, please revise them carefully;
- References should include the latest papers appropriately, such as:
DOI:10.1016/j.foodchem.2025.143948
DOI:10.1021/acs.jafc.5c02884
DOI:10.3390/foods14111873
Author Response
Reviewer 3, Comment 1:
- There are still a few writing errors in the text, please revise them carefully;
Response to Reviewer 3, Comment 1:
We would like to express our sincere appreciation for the reviewers' constructive comments. Following a careful re-examination of the manuscript, we identified and corrected several errors as pointed out. These revisions have been incorporated into the revised manuscript, and the changes are clearly marked in green highlight for ease of reference (see line 94, 98, 113, 139, 169, 270)
We hope that the updated manuscript meets the reviewers' expectations and contributes to the continued improvement of the work.
Reviewer 3, Comment 2:
- References should include the latest papers appropriately, such as:
DOI:10.1016/j.foodchem.2025.143948
DOI:10.1021/acs.jafc.5c02884
DOI:10.3390/foods14111873
Response to Reviewer 3, Comment 2:
We would like to express our sincere gratitude for your proactive suggestion of references to improve the quality of our manuscript. In response, we have incorporated the three recommended references into the gut microbiota section of the Discussion (see line 387). These additions have significantly enriched the manuscript and provided greater depth to our interpretation.
Thank you once again for your valuable contribution.

Reviewer 4 Report
Comments and Suggestions for Authors
I am very pleased with this revised version of the manuscript, I thank the authors for taking the time to address all my comments. I am happy with the manuscript in its present form.
Author Response

(The authors gave the same response as above.)

Reviewer 5 Report
Comments and Suggestions for Authors
Most textual issues have been fixed, and the scientific rationale is clear.
Author Response

(The authors gave the same response as above.)
